# MiniHack the Planet: A Sandbox for Open-Ended Reinforcement Learning Research

**Mikayel Samvelyan**[+!] **Robert Kirk**[!] **Vitaly Kurin**[#] **Jack Parker-Holder**[#] **Minqi Jiang**[+!]
**Eric Hambro**[+] **Fabio Petroni**[+] **Heinrich Küttler**[+] **Edward Grefenstette**[+!] **Tim Rocktäschel**[+!]

[+]Facebook AI Research [!]University College London [#]University of Oxford

{samvelyan,rockt}@fb.com

## Abstract

Progress in deep reinforcement learning (RL) is heavily driven by the availability of challenging benchmarks used for training agents. However, benchmarks that are widely adopted by the community are not explicitly designed for evaluating specific capabilities of RL methods. While there exist environments for assessing particular open problems in RL (such as exploration, transfer learning, unsupervised environment design, or even language-assisted RL), it is generally difficult to extend these to richer, more complex environments once research goes beyond proof-of-concept results. We present MiniHack, a powerful sandbox framework for easily designing novel RL environments.[1] MiniHack is a one-stop shop for RL experiments with environments ranging from small rooms to complex, procedurally generated worlds. By leveraging the full set of entities and environment dynamics from NetHack, one of the richest grid-based video games, MiniHack allows designing custom RL testbeds that are fast and convenient to use. With this sandbox framework, novel environments can be designed easily, either using a human-readable description language or a simple Python interface. In addition to a variety of RL tasks and baselines, MiniHack can wrap existing RL benchmarks and provide ways to seamlessly add additional complexity.

## 1   Introduction

Advancing deep reinforcement learning [RL, 52] methods goes hand in hand with developing challenging benchmarks for evaluating these methods. In particular, simulation environments like the Arcade Learning Environment [ALE, 6] and the MuJoCo physics simulator [54] have driven progress in model-free RL and continuous control respectively. However, after several years of sustained improvement, results in these environments have started to reach superhuman performance [18, 61, 3] while many open problems in RL remain [16, 27, 25]. To make further progress, novel challenging RL environments and testbeds are needed.

On one hand, there are popular RL environments such as Atari [7], StarCraft II [57], DotA 2 [41], Procgen [13], Obstacle Tower [29] and NetHack [33] that consist of entire games, but lack the ability to test specific components or open problems of RL methods in well-controlled proof-of-concept test cases. On the other hand, small-scale tightly controlled RL environments such as MiniGrid [12], DeepMind Lab [5], Alchemy [58], MetaWorld [60], and bsuite [43] have emerged that enable researchers to prototype their RL methods as well as to create custom environments to test specific open research problems (such as exploration, credit assignment, and memory) in

---

[1]Code available at https://github.com/facebookresearch/minihack

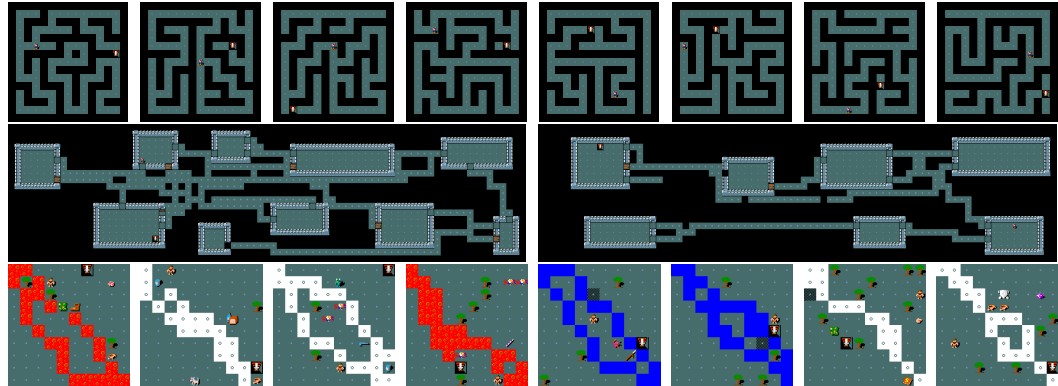

Figure 1: Examples of procedurally generated environments using the `des-file` format. **(Top)**: `MAZEWALK` command applied on a `15x15` grid, **(Middle)** corridors generated via `RANDOM_CORRIDOR`, **(Bottom)**: environments generated using the code snippet from Figure 2.

isolation. However, once specific research hypotheses are verified in these controllable simplified environments, RL practitioners find themselves between a rock and a hard place. Systematically extending such environments and gradually dropping simplifying assumptions can require arduous engineering and excessive time commitment, while opting for more challenging benchmarks [e.g. 33] often deprives researchers of a controllable path for assessing subsequent hypotheses. While frameworks like PyVGDL [48], GVGAI [44], and Griddly [4] can be used to design custom testbeds, creating complex environments with rich entities and environment dynamics would still require substantial engineering effort as complex environment interactions would have to be designed from scratch. Thus, there is a gap in terms of a framework that allows one to easily specify a suite of rich, gradually more difficult tasks, while also providing a large set of entities with complex environment interactions ready to be used to implement these tasks.

To fill this gap, we present MiniHack, a sandbox framework for easily designing novel RL environments and enriching existing ones. At the core of MiniHack are description files for defining procedurally generated worlds via the powerful domain-specific language (DSL) of the game of NetHack [47]. The full game of NetHack, arguably the richest gridworld benchmark in RL [33], is not suitable for answering specific research questions in isolation. However, NetHack's DSL allows MiniHack to tap into the richness of the game with its hundreds of pre-implemented entities and the complex interaction mechanics between them [38]. Furthermore, this DSL is flexible enough to easily build a wide range of testbeds, creating rich and diverse custom environments using only a few lines of human-readable code (see examples in Figure 1). Once written, either directly or using a convenient Python interface, MiniHack compiles the provided description files and wraps them as standard Gym environments [8].

The clear taxonomy of increasingly difficult tasks, the availability of multi-modal observations (symbolic, pixel-based, and textual), its speed and ease of use, make MiniHack an appealing framework for a variety of different RL problems. In particular, MiniHack could be used to make progress in areas such as unsupervised skill discovery, unsupervised environment design, transfer learning, and language-assisted RL. In addition to a broad range of environments that can easily be designed in the MiniHack framework, we also provide examples on how to import other popular RL benchmarks, such as MiniGrid [12] or Boxoban [21], to the MiniHack planet. Once ported, these environments can easily be extended by adding several layers of complexity from NetHack (e.g. monsters, objects, dungeon features, stochastic environment dynamics, etc) with only a few lines of code.

In order to get started with MiniHack environments, we provide a variety of baselines using frameworks such as TorchBeast [32] and RLlib [34], as well as best practices for benchmarking (see Section 3.5). Furthermore, we demonstrate how it is possible to use MiniHack for unsupervised environment design, with a demonstration of the recently proposed PAIRED algorithm [14]. Lastly, we provide baseline learning curves in Weights&Biases format[2] for all of our experiments and a detailed documentation of the framework.[3]

---

[2]https://wandb.ai/minihack
[3]https://minihack.readthedocs.io

In summary, this paper makes the following core contributions: (i) we present MiniHack, a sandbox RL framework that makes it easy for users to create new complex environments, (ii) we release a diverse suite of existing tasks, making it possible to test a variety of components of RL algorithms, with a wide range of complexity, (iii) we showcase MiniHack's ability to port existing gridworld environments and easily enrich them with additional challenges using concepts from NetHack, and (iv) we provide a set of baseline agents for testing a wide range of RL agent capabilities that are suitable for a variety of computational budgets.

## 2  Background: NetHack and the NetHack Learning Environment

The NetHack Learning Environment [`NLE`, 33] is a Gym interface [8] to the game of NetHack [47]. NetHack is among the oldest and most popular terminal-based games. In NetHack, players find themselves in randomly generated dungeons where they have to descend to the bottom of over 50 procedurally generated levels, retrieving a special object and thereafter escape the dungeon the way they came, overcoming five difficult final levels. Actions are taken in a turn-based fashion, and the game has many stochastic events (e.g. when attacking monsters). Despite the visual simplicity, NetHack is widely considered as one of the hardest games in history [53]. It often takes years for a human player to win the game for the first time despite consulting external knowledge sources, such as the NetHack Wiki [38]. The dynamics of the game require players to explore the dungeon, manage their resources, and learn about the many entities and their game mechanics. The full game of NetHack is beyond the capabilities of modern RL approaches [33].

`NLE`, which focuses on the full game of NetHack using the game's existing mechanisms for procedurally generating levels and dungeon topologies, makes it difficult for practitioners to answer specific research questions in isolation. In contrast, with MiniHack we present an extendable and rich sandbox framework for defining a variety of custom tasks while making use of NetHack's game assets and complex environment dynamics.

## 3  MiniHack

MiniHack is a powerful sandbox framework for easily designing novel RL environments. It not only provides a diverse suite of challenging tasks but is primarily built for easily designing new ones. The motivation behind MiniHack is to be able to perform RL experiments in a controlled setting while being able to increasingly scale the difficulty and complexity of the tasks by removing simplifying assumptions. To this end, MiniHack leverages the description file (`des-file`) format of NetHack and its level compiler (see Section 3.1), thereby enabling the creation of many challenging and diverse environments (see Section 3.4).

### 3.1  `des-file` format: A Domain Specific Language for Designing Environments

The `des-file` format [39] is a domain-specific language created by the developers of NetHack for describing the levels of the game. `des-files` are human-readable specifications of levels: distributions of grid layouts together with monsters, objects on the floor, environment features (e.g. walls, water, lava), etc. All of the levels in the full game of NetHack have pre-defined `des-files`. The `des-files` are compiled into binary using the NetHack level compiler, and MiniHack maps them to Gym environments.

Levels defined via `des-file` can be fairly rich, as the underlying programming language has support for variables, loops, conditional statements, as well as probability distributions. Crucially, it supports underspecified statements, such as generating a random monster or an object at a random location on the map. Furthermore, it features commands that procedurally generate diverse grid layouts in a single line. For example, the `MAZEWALK` command generates complex random mazes (see Figure 1 **Top**), while the `RANDOM_CORRIDORS` command connects all of the rooms in the dungeon level using procedurally generated corridors (see Figure 1 **Middle**). Figure 2 presents a `des-file` code snippet that procedurally generates diverse environment instances on a `10x10` grid, as presented in Figure 1 **Bottom**.

Figure 3 shows a `des-file` for a level with fixed, pre-defined map layout (lines 3-13). Here, the '`.`', '`+`', and '`S`' characters denote grid cells for floor, closed door, and secret door, respectively, while '`|`'

```
$river=TERRAIN:{'L','W','I'}
SHUFFLE:$river
LOOP [2] {
   TERRAIN:randline (0,0),(10,10),5,
       $river[0]
   MONSTER:random,random
}
REPLACE_TERRAIN:(0,0,10,10),'.','T',5%
STAIR:random,down
```

Figure 2: A sample code snippet in des-file format language. The $river variable is used to sample a terrain feature ('L' for lava, 'W' for water and 'I' for ice). The LOOP block draws two rivers via the randline command and places two random monsters at random locations. The REPLACE_TERRAIN commands replaces 5% of floors ('.') with trees ('T'). A stair down is added at random locations.

```
1  MAZE:"simple_maze",' '
2  GEOMETRY:center,center
3  MAP
4    --- --- ---
5    |.| |.| |.|
6  ---S---S---S---
7  |.......+.+...|
8  ---+-----.-----
9  |.......+.+...|
10 ---S---S---S---
11   |.| |.| |.|
12   --- --- ---
13 ENDMAP
14 LOOP [5] {
15    OBJECT:'%',random
16    TRAP:random,random
17 }
18 [10%]: GOLD: 100,random
19 MONSTER:('B',"bat"),(3,3)
```

Figure 3: A des-file example for a simple NetHack level.

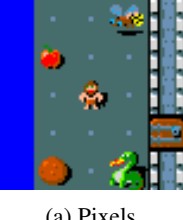

(a) Pixels

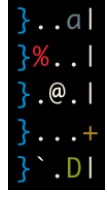

(b) Symbols

| water | floor | floor | killer bee | wall |
| water | an apple | floor | floor | wall |
| water | floor | **agent** | floor | wall |
| water | floor | floor | floor | closed door |
| water | a boulder | floor | green dragon | wall |

(c) Textual descriptions

Figure 4: Different forms of agent-centred observations of the grid of the map in MiniHack.

and '-' denote walls. The loop block (lines 14-17) places five random comestibles ('%') and random traps at random positions. Line 18 adds 100 golds at a random location with 10% probability. Line 19 adds a bat at a fixed location. These examples only provide a glimpse into the variety of levels that can be generated (see Appendix A for more examples and details on the des-file format).

### 3.2 MiniHack Environments

By tapping into the richness of the game of NetHack, MiniHack environments can make use of a large set of pre-existing assets. One can add one of more than 580 possible monster types, each of which has unique characteristics such as attack distance and type; health points; resistance against certain attacks; and special abilities such as changing shape, moving through walls, and teleporting. Practitioners can also choose from 450 items in the game, including various types of weapons, armour, tools, wands, scrolls, spellbooks, comestibles, potions, and more. These items can be used by the agent as well as monsters.

**Observations.** MiniHack supports several forms of observations, including global or agent-centred viewpoints (or both) of the grid (such as entity ids, characters, and colours), as well as textual messages, player statistics and inventory information [33]. In addition to existing observations in NLE, MiniHack also supports pixel-based observations, as well as text descriptions for all entities on the map (see Figure 4).

**Action Space.** NetHack has a large, structured and context-sensitive action space [47]. We give practitioners an easy way to restrict the action space in order to promote targeted skill discovery. For example, navigation tasks mostly require movement commands, and occasionally, kicking doors, searching or eating. Skill acquisition tasks, on the other hand, require interactions with objects, e.g. managing the inventory, casting spells, zapping wands, reading scrolls, eating comestibles, quaffing potions, etc. 75 actions are used in these tasks. A large number of actions and their nontrivial interactions with game objects offer additional opportunities for designing rich MiniHack tasks. For

example, a towel can be used as a blindfold (for protection from monsters that harm with their gaze), for wiping off slippery fingers (e.g. after eating deep-fried food from a tin), or even serve as a weapon when wet (which can be achieved by dipping the towel into water).

**Reward.** Reward functions in MiniHack can easily be configured. Our `RewardManager` provides a convenient way to specify one or more events that can provide different (positive or negative) rewards, and control which subsets of events are sufficient or required for episode termination (see Appendix B.4 for further details).

## 3.3 Interface

MiniHack uses the popular Gym interface [8] for the interactions between the agent and the environment. One way to implement MiniHack Gym environments is to write the description file in the human-readable `des-file` format and then pass it directly to MiniHack (see Figure 16a in Appendix B.2). However, users might find it more convenient to construct the environment directly in Python. Our `LevelGenerator` allows users to do this by providing the functionality to add monsters, objects, environment features, etc. Figure 5 presents an example code snippet of this process. Here, the agent starts near the entrance of a labyrinth and needs to reach its centre to eat the apple. A Minotaur, which is a powerful monster capable of instantly killing the agent in melee combat, is placed deep inside the labyrinth. There is a wand of death placed in a random location in the labyrinth. The agent needs to pick the wand up, and upon seeing the Minotaur, zap it in the direction of the monster. Once the Minotaur is killed, the agent would be able to reach the centre of the labyrinth and eat the apple to complete the task. The `RewardManager` is used to specify the goal that needs to be completed (eating an apple). Our `LevelGenerator` and `RewardManager` are described in more detail in Appendix B.2.

## 3.4 Tasks: A World of Possibilities

We release a collection of example environments that can be used to test various capabilities of RL agents, as well as serve as building blocks for researchers wishing to develop their own environments. All these environments are built using the interface described in Section 3.3, which demonstrates the flexibility and power of Mini-Hack for designing new environments.

```python
# Define the labyrinth as a string
grid = """
--------------------
|.......|.|........|
|.-----.|.|.-----|.|
|.|...|.|.|......|.|
|.|.|.|.|.|-----.|.|
|.|.|...|....|.|.|.|
|.|.--------.|.|.|.|
|.|.........|...|.|
|.|-------------|.|
|.................|
--------------------
"""
# Define a level generator
level = LevelGenerator(map=grid)
level.set_start_pos((9, 1))
# Add wand of death and apple
level.add_object("death", "/")
level.add_object("apple",
    place=(14, 5))
# Add a Minotaur at fixed position
level.add_monster(name="minotaur",
    place=(14, 6), args=("asleep",))

# Define the goal
reward_mngr = RewardManager()
reward_mngr.add_eat_event("apple")

# Declare task a Gym environment
env = gym.make(
    "MiniHack-Skill-Custom-v0",
    des_file=level.get_des(),
    reward_manager=reward_mngr)
```

Figure 5: A sample code snippet for creating a custom MiniHack task using the `LevelGenerator` and `RewardManager`.

**Navigation Tasks.** MiniHack's navigation tasks challenge the agent to reach the goal position by overcoming various difficulties on their way, such as fighting monsters in corridors (see Figure 6a), crossing a river by pushing boulders into it (see Figure 6b), navigating through complex or procedurally generated mazes (see Figure 1 **Top** and **Medium**). These tasks feature a relatively small action space.[4] Furthermore, they can be easily extended or adjusted with minimal effort by either changing their definition in Python or the corresponding `des-file`. For instance, once the initial version of the task is mastered, one can add different types of monsters, traps or dungeon features, or remove simplifying assumptions (such as having a fixed map layout or full observability), to further challenge RL methods. Our suite of 44 diverse environments is meant to assess several of the core capabilities of RL agents, such as exploration, planning, memory, and generalisation. The detailed

---

[4]Movement towards 8 compass directions, and based on the environment, search, kick, open, and eat actions.

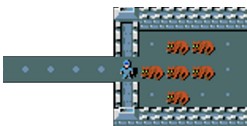

(a) `CorridorBattle` requires luring monsters into a corridor and fighting them one at a time.

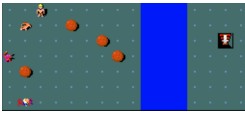

(b) `River` requires pushing boulders into a river to reach the goal via the generated bridge.

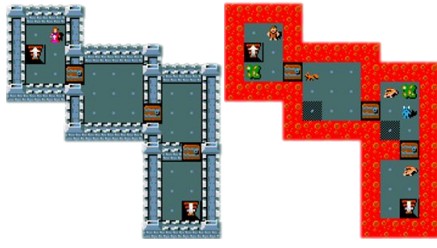

(c) Two versions of `MultiRoom-N4-S5` task. (left) Regular version (right) Extreme version that includes random monsters, locked doors, and lava tiles instead of walls.

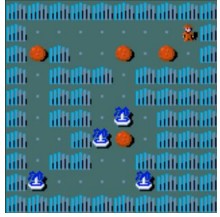

(d) `Boxoban` requires pushing boulders into different goals (here represented as four fountains).

Figure 6: Screenshots of several MiniHack tasks.

descriptions of all navigation tasks, as well as the full list of 44 registered environments, can be found in Appendix C.1.

**Skill Acquisition Tasks.** Our skill acquisition tasks enable utilising the rich diversity of NetHack objects, monsters and dungeon features, and the interactions between them. These tasks are different from navigation tasks in two ways. First, the skill acquisition tasks feature a large action space (75 actions), where the actions are instantiated differently depending on which object they are acting on. Given the large number of entities in MiniHack, usage of objects with an appropriate action in the setting of sparse rewards is extremely difficult, requiring a thorough exploration of the joint state-action space.[5] Second, certain actions in skill acquisition tasks are factorised autoregresively [45], i.e., require performing a sequence of follow-up actions for the initial action to have an effect. For example, to put on a ring, the agent needs to select the `PUTON` action, choose the ring from the inventory and select which hand to put it on. As a result, MiniHack allows getting rid of simplifying assumptions that many RL environments impose, such as having a single "schema" action used with a variety of objects regardless of the context. For the full list of tasks, see Appendix C.2.

**Porting Existing Environments to MiniHack.** Transitioning to using a new environment or benchmark for RL research can be troublesome as it becomes more difficult to compare with prior work that was evaluated on previous environments or benchmarks. Here, we show that prior benchmarks such as MiniGrid [12] and Boxoban [21] can be ported to MiniHack. While the MiniHack versions of these tasks are not visually identical to the originals, they still test the same capabilities as the original versions, which enables researchers familiar with the original tasks to easily analyse the behaviour of agents in these new tasks. Due to the flexibility and richness of MiniHack, we can incrementally add complexity to the levels in these previous benchmarks and assess the limits of current methods. This is especially useful for MiniGrid, where current methods are able to solve all existing tasks [20, 62].

As an example, we present how to patch the navigation tasks in MiniGrid [12] and increase their complexity by adding monsters, locked doors, lava tiles, etc (see Figure 6c). Similarly, we make use of publicly available levels of Boxoban [21] to offer these task in MiniHack (see Figure 6d). Once ported to MiniHack, these levels can easily be extended, for example, by adding monsters to fight while solving puzzles. An added benefit of porting such existing benchmarks is that they can use a common observation and action space. This enables investigating transfer learning and easily benchmarking a single algorithm or architecture on a wide variety of challenges, such as the planning problems present in Boxoban and the sparse-reward exploration challenges of MiniGrid.

While MiniHack has the ability to replace a large set of entities ported from original environments, it is worth noting that not all entities have identical replacements. For example, MiniGrid's `KeyCorridor` includes keys and doors of different colours, whereas the corresponding objects in NetHack have no colour. The randomly moving objects in MiniGrid's `Dynamic-Obstacles` tasks are also absent.

---

[5]Most of the state-of-the-art exploration methods, such as RND [9], RIDE [46], BeBold [62], and AGAC [20], rely on state space exploration rather than the state-action space exploration.

Despite MiniHack versions of ported environments having minor differences compared to originals, they nonetheless assess the exact same capabilities of RL agents. In particular, while the underlying dynamics of the environment are identical to the original in the case of Boxoban, our MiniGrid version includes slight changes to the agent's action space (turning and moving forwards vs only directional movement) and environment dynamics (the event of opening doors is probabilistic in MiniHack).

## 3.5 Evaluation Methodology

Here we describe the evaluation methodology and experimental practice we take in this paper in more detail, as a recommendation for future work evaluating methods on the MiniHack suite of tasks. To ensure a fair comparison between methods, performance should be evaluated in standard conditions. Specifically to MiniHack, this means using the same: action space,[6] observation keys,[7] fixed episode length, reward function, game character,[8] or any other environment parameter that can potentially affect the difficulty of tasks. When reporting results, we recommend reporting the median reward achieved over at least 5 independent training runs with different seeds. Reporting the median avoids the effects of any outliers, and five independent runs strike a balance between statistical validity and computational requirements. When evaluating the generalisation, agents should be trained on a limited number of seeds and tested on held-out seeds or the full distribution of levels. As well as sharing final performance, sharing full learning curves (provide all independent runs or present an error metric, such as standard deviation), wall-clock time and examples of behaviour are recommended as it can be helpful to other researchers building on or comparing against the results. If additional information has been used during training or testing, then that should be made clear in any comparisons with other work.

# 4 Experiments

In this section, we present experiments on tasks described in Section 3.4. The purpose of these experiments is to assess various capabilities of RL agents, and highlight how incremental changes in the environments using the rich entities in MiniHack further challenge existing RL methods. We highlight and discuss results on several different tasks, while results for all tasks can be found in Appendix E.

Our main baseline for all tasks is based on IMPALA [19], as implemented in TorchBeast [32]. For navigation tasks, we also use two popular exploration techniques – Random Network Distillation [RND, 9] and RIDE [46], the latter being designed specifically for procedurally generated environments where an agent is unlikely to visit a state more than once. We train and test agents on the full distribution of levels in each environment. We make use of the same agent architecture as in [33]. All details on agent architectures and training setting are described in Appendix D. Full results on all MiniHack tasks are available in Appendix E.

**Navigation Tasks.** Figure 7 summarises results on various challenging MiniHack navigation tasks. While simpler versions of the tasks are often quickly solved by the baseline approaches, adding layers of complexity (such as increasing the size of procedurally generated mazes, resorting to partially observable settings, and adding monsters and traps) renders the baselines incapable of making progress. For example, our baselines fail to get any reward on the most difficult version of the `River` task that includes moving monsters and deadly lava tiles, challenging the exploration, planning and generalisation capabilities of the agent. The results on the `KeyRoom` tasks highlight the inability of RL methods to handle generalisation at scale. Though the smaller version of the task (`KeyRoom-Random-S5`) is solved by all baselines, the larger variant (`KeyRoom-Random-S15`) is not solved by any of the methods.

---

[6]Larger action spaces can often increase the difficulty of the task.

[7]Abstractly, the same observation space. There are multiple (multimodal) options which may change the difficulty of the task.

[8]The default character in MiniHack is a chaotic human male rogue (`rog-hum-cha-mal`) for navigation tasks and a neutral human male caveman (`cav-hum-new-mal`) for skill acquisition tasks. For the `CorridorBattle` tasks, we override the default and use a lawful human female knight (`kni-hum-law-fem`) instead.

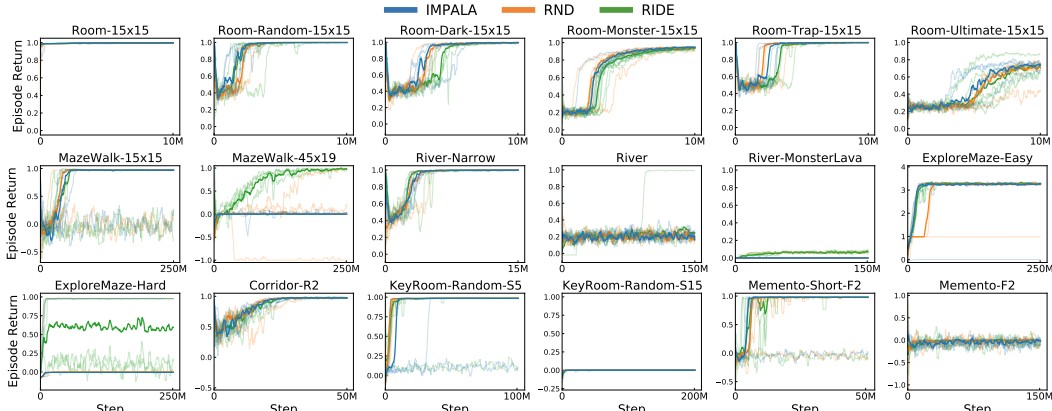

Figure 7: Mean episode returns on several MiniHack navigation tasks across five independent runs. The median of the runs is bolded.

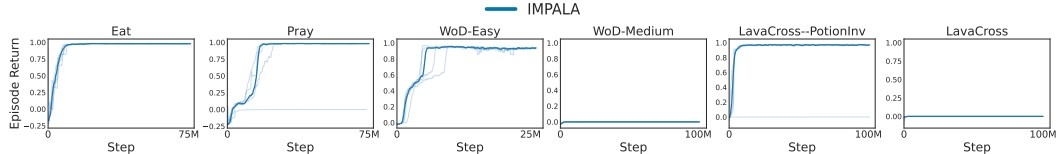

Figure 8: Mean episode returns on several skill acquisition tasks across five independent runs. The median of the runs is bolded.

**Skill Acquisition Tasks.** Figure 8 presents our results on various skill acquisition tasks. While the simple tasks that require discovering interaction between a single entity and an action of the agent (e.g., eating comestibles, zapping a wand of death towards a monster, etc.) can be solved by the baselines, the discovery of a sequence of entity-action relations in the presence of environment randomisation and distracting entities remains challenging for RL methods. For instance, none of the runs is able to make progress on `WoD-Medium` or `LavaCross` tasks due to insufficient state-action space exploration despite mastering the simplified versions of them.

**Ported Environments.** Figure 9 presents the results of different versions of the `MultiRoom` environment ported from MiniGrid [12]. In the version with two rooms, adding additional layers of complexity, such as locked doors, monsters, or lava tiles instead of walls, makes the learning more difficult compared to regular versions of the task. In the Extreme version with all the aforementioned complexities, there is no learning progress at all. In the version with four rooms, the baseline agents only make progress on the simplest version and the version with added monsters. All additional complexities are beyond the capabilities of baseline methods due to the hard exploration that they

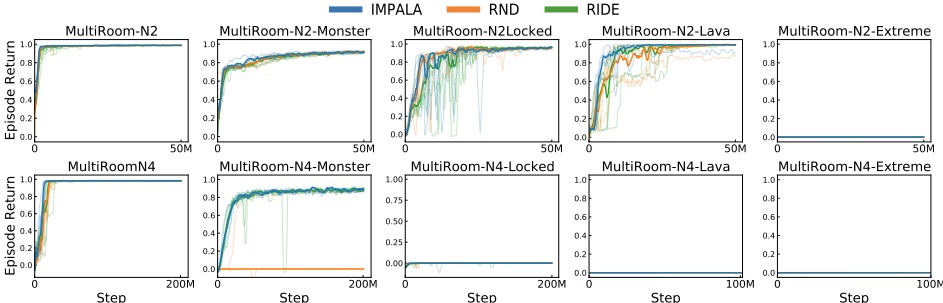

Figure 9: Mean episode returns on various `MultiRoom` environments ported from MiniGrid [12] across five independent runs. The median of the runs is bolded.

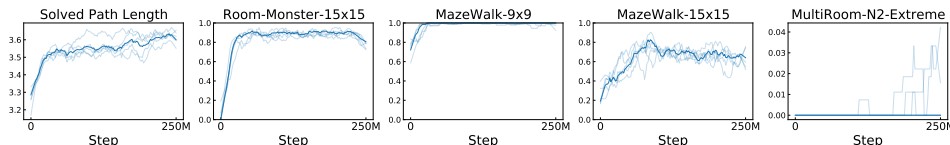

Figure 10: Results from the PAIRED algorithm, showing the solved path length of UED environments and zero-shot transfer performance. Plots show five independent runs with the median bolded.

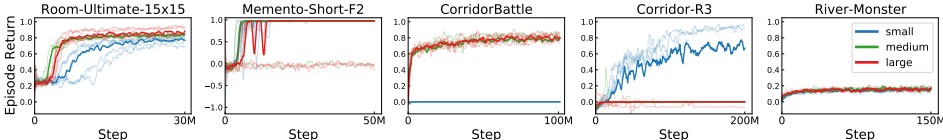

Figure 11: Mean episode returns of a baseline IMPALA agent using three model architectures.

require. Consequently, all independent runs fail to obtain any positive reward. These results highlight the ability of MiniHack to assess the limits of RL algorithms in an extendable, well-controlled setting.

**Unsupervised Environment Design.** MiniHack also enables research in *Unsupervised Environment Design* (UED), whereby an adaptive task distribution is learned during training by dynamically adjusting free parameters of the task MDP. MiniHack allows overriding the description file of the environment, making it easy to adjust the MDP configuration as required by UED. To test UED in MiniHack, we implement the recent PAIRED algorithm [14], which trains an environment adversary to generate environments in order to ultimately train a robust *protagonist* agent, by maximizing the regret, defined as the difference in return between a third, *antagonist* agent and the protagonist. We allow our adversary to place four objects in a small 5x5 room: {`walls`, `lava`, `monster`, `locked door`}. As a result of the curriculum induced by PAIRED, the protagonist is able to improve zero-shot performance on challenging out-of-distribution environments such as `MultiRoom-N2-Extreme`, despite only ever training on a much smaller environment (see Figure 10).

**Agent Architecture Comparison.** We perform additional experiments to compare the performance of the baseline IMPALA agent using different neural architectures. Figure 11 presents results using three architectures (small, medium, and large) on selected MiniHack tasks which differ in the number of convolutional layers, the size of hidden MLP layers, as well as the entity embedding dimension (see Appendix D.3 for full details). The performances of medium and large agent architectures are on par with each other across all five tasks. Interestingly, the small model demonstrates poor performance on `Room-Ultimate-15` and `CorridorBattle` environments, but outperforms larger models on the `Corridor-3` task. While it is known that small models can outperform larger models (both in terms of depth and width) depending on the complexity of the environment [1, 24], MiniHack opens door to investigate this phenomenon in a more controlled setting due to the generous environment customisation abilities it provides. We thus believe MiniHack would be a useful testbed for exploring the impact of architecture on RL agent training.

**RLlib Example.** To help kickstart the development of RL models using MiniHack, we also provide integration with RLlib [34]. RLlib enables using a wide range of RL algorithms within the framework, ensuring that research on MiniHack can be performed with varying computational constraints. Figure 12 presents the results of DQN [36], A2C [35], and PPO [51] methods on the `Room-15x15` task. See Appendix D.4 for more details.

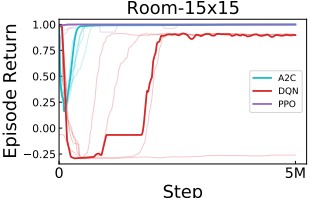

Figure 12: RLlib results.

## 5 Related Work

The RL community has made extensive use of video games as testbeds for RL algorithms [7, 57, 41, 40, 31]. However, such games are computationally expensive to run and not easily modifiable. Furthermore, they provide only a fixed set of levels, which results in the overfitting of trained policies. ProcGen [13] partially addresses this issue by providing a collection of 2D games with procedurally generated levels. However, the richness of games is still limited and only minor modifications are

possible, unlike the rich environment creation capabilities that MiniHack provides. Based on the C/C++ engines of NetHack and `NLE`, MiniHack is 16 times faster than ALE [7] and faster than ProcGen by approximately 10% (see Appendix D of [33] for an approximate comparison).

MiniGrid [12] addresses the issue of computational efficiency by providing a collection of procedurally generated grid-world tasks. Nevertheless, the complexity of the environments is still limited, containing only a few types of entities and small action space. Moreover, extending the existing environments is difficult as it requires understanding the internals of the library. MiniHack provides a much richer set of entities (hundreds of objects, monsters, dungeon features) and a much larger action space. Moreover, MiniHack is designed to be easy to extend and build on top of, only requiring familiarity with a flexible and expressive high-level DSL but no underlying implementation details.

*bsuite* [43] features a set of simple environments designed to test specific capabilities of RL agents, such as memory or exploration. In contrast, MiniHack is not confined to a static task set allowing researchers to easily extend the existing task set without the need to understand the implementation details of the framework.

Several roguelike games, a genre of video games characterised by progressing through procedurally generated dungeon levels and grid-based movements, have been proposed as RL benchmarks. Rogueinabox [2] provides an interface to Rogue, a roguelike game with simple dynamics and limited complexity. Rogue-Gym [30] introduces a simple roguelike game built for evaluating generalisation in RL and uses parameterisable generation of game levels. `NLE` [33] and *gym-nethack* [10, 11] provide a Gym interface around the game of NetHack. However, these benchmarks use either fixed, predefined level descriptions of the full games, or a fixed set of concrete subtask (e.g., 1-on-1 combat with individual monsters [11]). In contrast, MiniHack allows easily customising dungeon layouts and placement of environment features, monsters and objects by a convenient Python interface or a human-readable description language.

MiniHack is not the first to provide a sandbox for developing environments. PyVGDL [48, 49] uses a concrete realisation of the Video Game Description Language [VGDL, 17] for creating 2D video games. The original software library of PyVGDL is no longer supported, while the 2.0 version is under development [56]. The GVGAI framework [44] is also based on the VGDL but suffers from a computational overhead due to its Java implementation. Griddly [4] provides a highly configurable mechanism for designing diverse environments using a custom description language. Similar to MiniHack, Griddly is based on an efficient C/C++ core engine and is fast to run experiments on. Griddly is approximately an order of magnitude faster than MiniHack for simple environments, but it is unclear to what extent adding complex dynamics to Griddly, equivalent to what MiniHack provides, will decrease its speed. Furthermore, Griddly supports multi-agent and real-time strategy (RTS) games, unlike MiniHack. While PyVGDL, GVGAI, and Griddly can be used to create various entities, developing rich environment dynamics requires a significant amount of work. In contrast, MiniHack features a large collection of predefined objects, items, monsters, environment features, spells, etc and complex environment mechanics from the game of NetHack, thus hitting the sweet spot between customizability and the richness of entities and environment dynamics to draw upon. MiniHack also provides a rich set of multimodal observations (textual descriptions, symbolic and pixel-based observations) and a convenient Python interface for describing environments in only a few lines of code. Finally, the Malmo Platform [28] and MineRL [23] provides an interface to a popular game of Minecraft. While being rich in the environment dynamics, Minecraft is computationally intensive compared to NetHack [47] (MiniHack is approximately 240 times faster than MineRL [33]).

## 6   Conclusion

In this work, we presented MiniHack, an easy-to-use framework for creating rich and varied RL environments, as well as a suite of tasks developed using this framework. Built upon NLE and the `des-file` format, MiniHack enables the use of rich entities and dynamics from the game of NetHack to create a large variety of RL environments for targeted experimentation, while also allowing painless scaling-up of the difficulty of existing environments. MiniHack's environments are procedurally generated by default, ensuring the evaluation of systematic generalization of RL agents. The suite of tasks we release with MiniHack tests the limits of RL methods and enables researchers to test a wide variety of capabilities in a unified experimental setting. To enable further experimentation and research, we open-source our code for training agents with numerous RL algorithms, outline best practices on how to evaluate algorithms on the benchmark suite of tasks, and provide baseline results on all of the tasks that we released.

## Acknowledgements

We thank Danielle Rothermel, Zhengyao Jiang, Pasquale Minervini, Vegard Mella, Olivier Teytaud, and Luis Pineda for insightful discussions and valuable feedback on this work. We also thank our anonymous reviewers for their recommendations on improving this paper.

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
