# OpenReview forum: "MiniHack the Planet: A Sandbox for Open-Ended Reinforcement Learning Research"
_NeurIPS.cc/2021/Track/Datasets_and_Benchmarks/Round1 — NeurIPS 2021 Datasets and Benchmarks Track (Round 1)_

### Official Review · Reviewer_7K3G · 2021-07-04
**Interesting and useful dataset**

**Rating:** 7
**Confidence:** 3

**Strengths:**

+ The dataset is interesting and relevant. In my opinion, it fills an interesting gap in data generation for RL, showing a good balance between the capability of creating complex scenarios for several tasks, the effort needed to do it, and the amount of configurable features.
+ The documentation contains ample technical details and examples of use, which potentially will allow an easy use in the initial stages.
+ The authors made an effort to port existing scenarios, which will facilitate the adoption of the tool.

**Weaknesses:**

- The existing environments that have been ported are not identical. This is a limitation, as new results using MiniHack could not be comparable to existing ones. The authors should be more specific on this and state which tasks would be comparable with existing results and which ones not. Would there be any chance that the ported environments look identical?

- I think the evaluation methodology section, which is in the supplementary material right now, is quite relevant for the framework use. Wouldn't it be better to move it to the main paper?

**Additional Feedback:**

In general, I have a positive view of this dataset. I would encourage the authors to work on the few weaknesses I found. Most importantly, if I am not wrong, there is not maintenance plan for MiniHack, and this is a critical aspect. The authors should design one carefully. The second critical aspect to ensure a wide adoption of the framework is its compatibility with previous scenarios. The authors should be more specific about the compatibility of the existing ones, and they should make an effort to make existing scenarios identical (if possible).

**Clarity:**

The paper is well written and clear. As a personal opinion, the general tone is a bit enthusiastic (e.g., powerful is repeated in excess), I prefer manuscripts that are written in a more neutral tone.

**Correctness:**

The paper is correct as far as I checked. The framework and the explanations are sufficiently detailed. The baselines and metrics shown in the tests of the framework are correct.

**Documentation:**

The level of detail on technical aspects is appropriate. The URL was provided and the dataset is easy to access and download. As the only weakness here, I could not find a maintenance plan, which is essential to guarantee the use of the framework over the years.

**Ethics:**

Not applicable.

**Relation To Prior Work:**

The previous work is nicely covered up to my knowledge.

**Summary And Contributions:**

The paper presents MiniHack, a framework for creating RL environments that leverages the tools and procedures of the NetHack game. MiniHack is motivated by the need of creating a continuum in the dataset space, which would allow to evaluate algorithms in scenarios of gradually increasing difficulty. Popular RL environments have been ported to MiniHack. The framework is tested in navigation and skill acquisition tasks using reasonable recent baselines, where it is shown how gradual changes in some data features change the difficulty of the scenarios, sometimes challenging existing baselines.

---

> ### Author Response · Authors · 2021-07-08
> **Responses to the review**
>
> We thank the reviewer for their feedback that we used to improve our work. It is great to hear that the reviewer found that MiniHack strikes a good balance between customisability, ease of use and complexity, as that's something we aimed to create with this benchmark. We're also glad the reviewer found the documentation useful. We address your main concerns below, and hope that this could lead to you considering strengthening your recommendation or explaining what still stands in the way, so that we may further improve the paper.
>
> ### On the evaluation methodology section
>
> Thank you for the suggestion. We have moved the evaluation methodology section to the main part of the paper. Please have a look at the updated PDF.
>
> ### On the maintenance plan
>
> Thank you for bringing this up. We agree this is extremely important, and we have already added the maintenance plan to the public repository following your comment, which can be found [here](https://github.com/MiniHackPlanet/MiniHack/blob/master/MAINTENANCE.md). Our maintenance plan follows a widely-adopted template for datasets and benchmarks proposed by [(Gebru et al., 2020)](https://arxiv.org/abs/1803.09010). We will publicly be releasing MiniHack as open source on GitHub under a permissive license, meaning that in contrast with large datasets, the resource will be available as long as Github hosts it (hopefully in perpetuity), and that any group or individual will have the ability to contribute to or even take over (e.g. via forking) the maintenance of the project if for some reason the authors cannot provide continuity of support. That said, the designers of the environment are a diverse group spanning several industrial and academic labs, providing some degree of robustness to the short-to-midterm support we hope to offer: if one lab's priorities shift in a way which endanger its support of the environment, it is not necessary that the others will too, and thus we have a better chance of offering continuity of support. To further assist the user of MiniHack, we provide detailed instructions with pre-defined [GitHub templates for bug reports and features requests](https://github.com/MiniHackPlanet/MiniHack/issues/new/choose), as well as [instructions on how to contribute to MiniHack.](https://github.com/MiniHackPlanet/MiniHack/blob/master/CONTRIBUTING.md)
>
> ### On differences of ported environments
>
> It is not our goal that the ported environments are identical to existing ones. Our primary objective is to provide practitioners a means to implement much richer and more interesting tasks by adding additional complexities to ported environments that MiniHack provides, therefore the 1-to-1 comparability is of secondary importance. An added benefit of porting such existing benchmarks is that they can use a common observation and action space. This enables investigating transfer learning and easily benchmarking a single algorithm or architecture on a wide variety of challenges, such as the planning problems present in Boxoban and the sparse-reward exploration challenges of MiniGrid.
>
> However, we note that MiniHack versions of these tasks still test the same capabilities as the original versions (such as exploration and planning). In fact, in the case of Boxoban, the underlying dynamics of the environment are identical to the original, while our MiniGrid version includes only slight differences to the agent's action space (turning and moving forwards vs just directional movement) and environment dynamics (the event of opening doors is probabilistic in MiniHack). Furthermore, MiniHack can additionally provide multi-modal observations (e.g. pixel-based, symbolic, and textual information about the grid), which is not supported in the original environments.
>
> Following your comment, we added additional information in the updated manuscript highlighting the differences between original benchmarks and ported variants to avoid any confusion.

---

### Official Review · Reviewer_PUtC · 2021-07-05
**Interesting work, but seems to be an incremental enhancement to NetHack environment**

**Rating:** 7
**Confidence:** 4
**Correctness:** No concern.

**Strengths:**

1.	MiniHack introduces many new environments and ways to create new environments. The description file in MiniHack seems very general and could be used to create various environments. I believe MiniGrid could help facilitate future RL research and would be of interest to many RL researchers.
2.	The interfaces seem to be easy to use. The code structure and the README are easy to follow. The authors interface MiniGrid with RLlib and Torchbeast, which could help users quickly run experiments.

**Weaknesses:**

1.	MiniHack seems to be a very incremental enhancement to NetHack. Specifically, the codebase seems to generally follow NetHack. The only difference of this paper is to introduce des-file to customize the environment. It is unclear to me why this new feature cannot be directly integrated into the NetHack environment. While I like the motivation and the idea of MiniHack, this new feature seems to be not significant enough as a separate contribution.
2.	The benchmark experiments seem weak. Recently, many new algorithms are proposed to address procedurally-generated environments. Since MiniHack provides procedurally-generated environments, in addition to IMPALA and RND, I would expect some results using algorithms designed for procedurally-generated environments. The experiments could also be enhanced by trying different configurations, such as neural architectures.

**Additional Feedback:**

See strengths and weaknesses.

**Clarity:**

The paper is clear with detailed descriptions of the motivation, interfaces, etc.

**Documentation:**

The README is easy to follow. The authors have also provided dockers for easily setting up environments.

**Ethics:**

NaN

**Relation To Prior Work:**

The authors have discussed various previous environments as well as sandboxes. While MiniHack is the not first sandbox, it is fast and can provide rich environmental dynamics.

**Summary And Contributions:**

The paper presents MiniHack, a package for easily designing new RL environments to study different challenges in RL. MiniHack is built upon their previous environment NetHack. The authors argue that the full game of NetHack is too challenging, and thus it is not suitable for answering specific research questions in isolation. Motivated by this, MiniHack allows creating different environments by modifying des-file, which is a domain-specific language to describe the environment. Then the authors create various environments as well as porting some exiting environments to MiniHack. Finally, the authors present benchmark results with IMPALA and RND, as well as some algorithms in RLlib.

---

> ### Author Response · Authors · 2021-07-08
> **Responses to the review (Part 1)**
>
> We thank the reviewer for their feedback that we will use to improve our paper. It is great to hear that the reviewer found that MiniHack could help facilitate future RL research and that the interfaces that we provide are easy-to-use and helpful. We address your main concerns below and hope that this will lead to you considering strengthening your recommendation or explaining what still stands in the way, so that we may further improve the paper.
>
> ### MiniHack is not an incremental enhancement to NetHack
>
> While MiniHack does use the NetHack Learning Environment [(NLE, Küttler et al, 2020)](https://arxiv.org/abs/2006.13760) to communicate with the NetHack game, it is a conceptually different benchmark designed for different use cases. MiniHack provides a sandbox framework for easily designing rich and diverse RL environments for specific open research questions (such as exploration, skill discovery/transfer, unsupervised environment design, etc), whereas NLE proposes the full game of NetHack as a grand challenge for RL.
>
> The [refactored codebase of MiniHack](https://github.com/MiniHackPlanet/MiniHack), which is now entirely decoupled from NLE and is a separate Python package, additionally adds the following functionalities that are not present in NLE:
>
> 1. The ability to customise the description files and make such rich and general environments based on the des-file format was not included in the original NLE codebase. Considerable engineering effort was required for making this possible quickly and efficiently, including changes to the core NetHack source code.
> 2. The observation space in MiniHack is much richer. NLE provides only symbolic observations (chars, colours, entity ids) about the grid, whereas MiniHack additionally provides pixel-based and textual observations.
> 3. MiniHack allows for a very customisable reward function using a convenient RewardManager interface. Users can specify specific goals, subgoals and penalties, such as eating certain comestibles or wearing certain armour, whereas NLE allows changing only the scale of the signal.
> 4. The LevelGenerator interface of MiniHack allows one to write automated description files using a convenient Python interface without the need to know specifics of the des-file format language, making level generation much more accessible than in the original NetHack source code.
> 5. MiniHack also includes an interface to the [NetHack wiki](http://nethackwiki.com/), an external knowledge source that can be used for language-assisted RL. Additionally, users can have immediate access to textual information about entities they observe on the grid.
> 6. We provide a large collection of 81 different tasks, specifically designed for assessing the core capabilities of RL agents in separation, such as exploration, memory, and planning.
> 7. MiniHack allows to easily port existing grid-based environments, such MiniGrid and Boxoban, and enhance them using the rich set of entities from the game.
> 8. MiniHack also opens the door for research in the field of Unsupervised Environment Design (see Section 4), which requires constantly changing the environment. As NLE only works with the fixed (randomised) environment that is the full game of NetHack, this would also not be possible with NLE codebase without the changes that programmatic environment creation framework that MiniHack provides.

---

> ### Author Response · Authors · 2021-07-08
> **Responses to the review (Part 2)**
>
> ### Choices of baseline agents
>
> We would be happy to run additional baseline experiments that can provide more insight into the complexities of existing tasks in MiniHack. We would like to know which methods specifically designed for procedurally generated environments the reviewer had in mind.
>
> We plan to include RIDE [(Raileanu and Rocktäschel, 2019)](https://openreview.net/forum?id=rkg-TJBFPB) as an additional baseline which is specifically designed for the procedurally generated environments. This should be included by the end of the review week and we believe will provide a more rigorous set of experiments.
>
> We also note that some of the recent RL approaches that perform well on procedurally generated environments, such as BeBold [(Zhang et al, 2020)](https://arxiv.org/abs/2012.08621), AMIGo [(Campero et al, 2020)](https://openreview.net/forum?id=ETBc_MIMgoX) and AGAC [(Flet-Berliac et al., 2021)](https://openreview.net/forum?id=_mQp5cr_iNy) are tailored specifically for MiniGrid on which they are evaluated. Specifically, all of these approaches assume exploration is mainly about discovering new (X, Y) positions due to the strong navigation challenge of MiniGrid. In contrast, MiniHack comes with a rich set of skill acquisition tasks that do not necessarily make visible changes on the grid, such as fighting monsters, utilising objects, and employing various skills. In addition, these previous methods concentrate on state-space exploration rather than state-action space exploration, which is needed for MiniHack's skill acquisition tasks. Thus, we believe it is reasonable to assume that the approaches above have overfitted to the specifics of MiniGrid and that they would not work well for most MiniHack tasks. However, we plan to verify this hypothesis empirically and update the repository and paper accordingly.
>
> We additionally provide baseline implementation and results of the recent PAIRED algorithm [(Dennis et al, 2020)](https://arxiv.org/abs/2012.02096) for _Unsupervised Environment Design_, which trains an adversary to generate environments in order to ultimately train a robust protagonist agent, by maximising the regret, defined as the difference in return between a third, antagonist agent and the protagonist. As a result of the curriculum induced by PAIRED, the protagonist is able to improve zero-shot performance on out-of-distribution environments such as MiniHack's MazeWalk environments.
>
> Furthermore, we hope that other members of the community will make use of MiniHack for benchmarking purposes and we plan to maintain a leaderboard of all methods on our repository.
>
> Regarding experiments using different neural architectures, we would like to know what specifically the reviewer had in mind. We report results using the configuration that yielded the strongest empirical results across various configurations that we tried. By providing MiniHack as an accessible benchmark for the community, we believe that the community will propose various configuration choices and novel neural architectures for handling the challenges that MiniHack offers.

---

> ### Comment · Reviewer_PUtC · 2021-07-11
> **Thanks for the reply from the authors**
>
> Thank you for the detailed reply. The response has addressed many of my concerns. I agree that MiniHack is a non-trial effort compare the NetHack. I also appreciate that the authors will add the results of RIDE. For the question of neural architecture, I would expect some results using shallow and deep architectures. Since procedurally generated have a very large observation space, a deeper network could lead to better performance. Some results of this aspect could be of interest to the users of the library. I will raise my score (assuming that the RIDE results can be added in the review period. I will check!)

---

> > ### Author Response · Authors · 2021-07-14
> > **Response to reviewer's comment**
> >
> > We thank the reviewer for providing an additional comment and clarification, which we used to improve the paper. We also thank the reviewer for indicating to raise the score once RIDE results are in place. Here are the changes we made to the updated manuscript:
> >
> > ### RIDE baseline results
> >
> > We added results of the RIDE algorithm [(Raileanu and Rocktäschel, 2019)](https://openreview.net/forum?id=rkg-TJBFPB) — an exploration technique designed specifically for procedurally generated environments — for all MiniHack navigation and ported tasks. In Figures 7 and 30 we observe that the RIDE baseline significantly outperforms both IMPALA and RND agents on 4 challenging tasks, namely MazeWalk-45x19, MazeWalk-Mapped-45x19, ExploreMaze-Hard and ExploreMaze-Hard-Mapped. All of these tasks feature large, procedurally generated mazes that RIDE agents learned to solve successfully due to the impact-driven intrinsic reward.
> >
> > ### Additional experiments with various agent architectures
> >
> > As per the reviewer's suggestion, we performed additional experiments to investigate how changes in agent architecture, particularly model depth, width and entity embedding size, affect the performance of the baseline agent. We experimented with three agent architectures on selected MiniHack tasks, which are added to Section 4 of the paper. The number of CNN layers, MLP hidden dimension size and entity embedding size are (3, 64, 16) for the *small* model, (5, 256, 64) for the *medium* model and (9, 512, 128) for the *large* model, respectively. This experiment provided interesting results, where the small model significantly outperformed larger models on one of the tasks (Corridor-R3). We thank the reviewer for suggesting us to perform this experiment. While it is known that small models can outperform larger models (both in terms of depth and width) depending on the complexity of the environment ([Andrychowicz et al, 2021](https://openreview.net/forum?id=nIAxjsniDzg), [Henderson et al, 2017](https://arxiv.org/abs/1709.06560)), MiniHack opens door to investigate this phenomenon in a more controlled setting due to the generous environment customisation abilities it provides. We thus believe MiniHack would be a useful testbed for exploring the impact of architecture on RL agent training.

---

> > > ### Comment · Reviewer_PUtC · 2021-07-15
> > > **Thanks for the additional experiments**
> > >
> > > The new results are quite interesting, particularly large/small networks seem to perform well in different environments. It is quite interesting. I believe MiniHack will motivate many future studies. I hereby raise my score to Accept.

---

### Official Review · Reviewer_Lxv7 · 2021-07-05

**Rating:** 6
**Confidence:** 3
**Correctness:** Yes
**Clarity:** Yes

**Strengths:**

The paper presents an easy-to-use RL benchmark that uses the popular OpenAI Gym interface. The environment can be created easily via either writing a description file or programmatically using Python. The proposed benchmark MiniHack also enables creation of many complex environments that can test the limit of RL algorithms in exploration, generalization and skill discovery. I think such a benchmark is of reasonable amount of significance to the RL community and also make RL environments more accessible to the community.

**Weaknesses:**

The proposed environment is still based on 2D games, which is less realistic and unclear if it would be of great impact on real-world RL applications. I'm also not sure if MiniHack would provide more interesting problems to study compared to existing benchmarks such as Atari. In general, it is unclear to me if the RL community requires more game-based benchmarks to test RL algorithms. I feel like testing RL algorithms in real-world situations are more appealing and important.

Furthermore, I think the evaluations of existing RL algorithms on MiniHack are not thorough. The authors should also evaluate more algorithms such as TD3, SAC, etc.

**Additional Feedback:**

see above.

**Documentation:**

Yes

**Relation To Prior Work:**

Yes

**Summary And Contributions:**

This paper presents a new benchmark on game-based RL environments that can be procedurally generated easily by users. The authors release a variety of tasks in the benchmark based on NetHack, making it possible to test various aspects of RL algorithms, such as navigation, skill acquisition and unsupervised environment design. The authors evaluated several baselines such as A2C, PPO, DQN, IMPALA and RND, on the proposed benchmark and show that there's room for improvement in future work.

---

> ### Author Response · Authors · 2021-07-08
> **Responses to the review (Part 1)**
>
> We thank the reviewer for their feedback that we will use to improve our paper. It is great to hear that the reviewer found MiniHack to be a significant contribution for the RL community in terms of accessibility, and our paper and benchmark documentation are clear and easy to follow. We address your main concerns below, and hope that this will lead to you considering strengthening your recommendation or explaining what still stands in the way, so that we may further improve the paper.
>
> ### On game-based benchmarks for RL
>
> While realistic 3D testbeds are clearly a desirable end goal for RL, *visually simplistic* environments (including 2D grid-based environments, or even tasks with symbolic rather than or as well as complex visual representations) play a crucial role in RL research ([Chevalier-Boisvert et al, 2018](https://github.com/maximecb/gym-minigrid), [Wang et al, 2021](https://arxiv.org/abs/2102.02926), [Young et al, 2019](https://arxiv.org/abs/1903.03176), [Silver et al, 2017a](https://arxiv.org/abs/1712.01815), [Silver et al, 2018b](https://www.nature.com/articles/nature24270), [Vinyals et al, 2019](https://www.nature.com/articles/s41586-019-1724-z)). Not only does this allow researchers to focus computational resources on isolated problems such as credit assignment, skill acquisition or exploration with sparse rewards, but it also facilitates more accessible research without the cost of processing pixel based observations. Furthermore, many real word-applications of RL are visually simple, such as energy usage optimisation [(Evans & Gao, 2016)](https://deepmind.com/blog/article/deepmind-ai-reduces-google-data-centre-cooling-bill-40), compiler optimisation [(Cummins et al, 2020)](https://github.com/facebookresearch/CompilerGym), network congestion control [(Sivakumar et al, 2019)](https://arxiv.org/abs/1910.04054) or chip design [(Mirhoseini et al, 2021)](https://www.nature.com/articles/s41586-021-03544-w). MiniHack allows us to deal with some of the many open problems in RL in separation by providing a means to design environments very quickly and incrementally increasing their difficulty. Once research hypotheses are verified on such convenient frameworks with superior runtime performance and low resource requirements, practitioners can move to more visually harder tasks. Finally, we note the significant use of MiniGrid in many of the latest RL research papers published in top venues ([https://github.com/maximecb/gym-minigrid](https://github.com/maximecb/gym-minigrid)). Clearly there is significant demand and use for this type of environment, and MiniHack adds several additional dimensions of complexity while crucially increasing customisability.
>
> ### Comparison to existing benchmarks such as Atari
>
> We appreciate that the community does not need more environments like Atari. That is precisely the motivation for MiniHack, which is more complex in a multitude of different ways:
>
> - Atari environments are so-called singleton environments. Every episode always looks the same, effectively training agents on the test set. This does not allow us to assess test-time generalisation abilities of the agent to unseen situations, which is crucial for real-world applications of RL. MiniHack environments, on the other hand, rely on procedural content generation and challenge the agent to handle previously unseen situations.
> - Atari environments are fixed. Being composed of entire games, Atari tasks cannot be easily modified. MiniHack environments, on the other hand, can be easily changed to accompany various assumptions that the practitioner might have, for example testing specific research hypotheses and modelling assumptions.
> - Atari is deterministic. Therefore, it is possible to achieve high scores by simply memorising a  sequence of actions rather than learning to make good decisions (see [Ecoffet et al, 2019](https://arxiv.org/abs/1901.10995)). This would not be possible in MiniHack environments, which are highly stochastic since many events, such as damage dealt to agent and monsters, spells, etc, are probabilistic by nature.
> - RL agents already perform at superhuman level on all Atari tasks ([Ecoffet et al, 2019](https://arxiv.org/abs/1901.10995), [Badia et al, 2020](https://arxiv.org/abs/2003.13350)), whereas in many MiniHack environments agents are unable to make meaningful progress, highlighting the complexity of MiniHack despite its visual simplicity.

---

> ### Author Response · Authors · 2021-07-08
> **Responses to the review (Part 2)**
>
> ### On additional baseline results
>
> - We appreciate the comment that it would be interesting to include additional baseline results and we are happy to run more algorithms that can provide more insight into the complexities of existing tasks in MiniHack. We plan to include an additional baseline RIDE [(Raileanu and Rocktäschel, 2019)](https://openreview.net/forum?id=rkg-TJBFPB) which was specifically designed for procedurally generated environments. This should be included by the end of the review week and we believe will provide a more rigorous set of experiments. We also plan to evaluate the effectiveness of MiniHack environments using recent exploration techniques, such as BeBold [(Zhang et al, 2020)](https://arxiv.org/abs/2012.08621), AMIGo [(Campero et al, 2020)](https://openreview.net/forum?id=ETBc_MIMgoX) and AGAC [(Flet-Berliac et al., 2021)](https://openreview.net/forum?id=_mQp5cr_iNy).
> - We do not plan to include TD3 and SAC given they are designed for continuous action spaces, whereas MiniHack environments use a discrete action-space. We note however that we include an RLlib integration, which means anyone can get started running [different algorithms](https://docs.ray.io/en/master/rllib-algorithms.html) in MiniHack.
> - Furthermore, we hope that other members of the community will make use of MiniHack for benchmarking purposes and we plan to maintain a leaderboard of all methods on our repository.

---

> ### Author Response · Authors · 2021-07-15
> **Additional baseline results**
>
> We would like to let the reviewer know that we added results of an additional algorithm, namely RIDE [(Raileanu and Rocktäschel, 2019)](https://openreview.net/forum?id=rkg-TJBFPB) which is an exploration method designed specifically for procedurally generated environments. In Figures 7 and 30 we observe that the RIDE baseline significantly outperforms both IMPALA and RND agents on 4 challenging tasks, namely MazeWalk-45x19, MazeWalk-Mapped-45x19, ExploreMaze-Hard and ExploreMaze-Hard-Mapped. All of these tasks feature large, procedurally generated mazes that RIDE agents learned to solve successfully due to the impact-driven intrinsic reward.
>
> We also extended the empirical evaluation of the paper by performing experiments using various neural architectures. Please see all updates to the paper in our joint response [here](https://openreview.net/forum?id=skFwlyefkWJ&noteId=fuEGyxYWzN).

---

### Author Response · Authors · 2021-07-14
**Joint response to all reviewers**

We thank the reviewers for their time and feedback. We are confident that in responding to individual questions and comments, the paper has already improved significantly. We have prepared an updated version of the paper and codebase, which we believe addresses all of the major points raised by the reviewers. Below we summarise the main improvements and additional results that we have added to our revision. In addition, we would like to address some misunderstandings that concerned some reviews.

### Additional baseline results

We thank the Reviewer PUtC for suggesting to add a new method to the list of baselines we evaluate MiniHack on. In particular, we added results of the RIDE algorithm [(Raileanu and Rocktäschel, 2019)](https://openreview.net/forum?id=rkg-TJBFPB) — an exploration technique designed specifically for procedurally generated environments — for all MiniHack navigation and ported tasks. In Figures 7 and 30 we observe that the RIDE baseline significantly outperforms both IMPALA and RND agents on several tasks that feature large, procedurally generated mazes which RIDE agents learned to solve successfully due to the impact-driven intrinsic reward.

### Additional experiments with various agent architectures

We are grateful to the Reviewer PUtC for advising us to perform additional experiments using various neural architectures. We provided additional results that investigate how changes in agent architecture, particularly model depth, width and entity embedding size, affect the performance of the baseline agent. The new results in Section 4 provide additional insight regarding neural architecture choices across various tasks.

### Maintenance plan

We thank Reviewer 7K3G for this great suggestion. We have added the maintenance plan to our public repository that follows a widely adopted template for datasets and benchmarks proposed by [(Gebru et al., 2020)](https://arxiv.org/abs/1803.09010) which can be found [here](https://github.com/MiniHackPlanet/MiniHack/blob/master/MAINTENANCE.md). To further assist the users of MiniHack, we provide detailed instructions with pre-defined templates for [bug reports and features requests](https://github.com/MiniHackPlanet/MiniHack/issues/new/choose), as well as [instructions](https://github.com/MiniHackPlanet/MiniHack/blob/master/CONTRIBUTING.md) on how to contribute to MiniHack.

### Evaluation Methodology

Following a recommendation by Reviewer 7K3G, we have moved the evaluation methodology section from the appendix to the main paper. We thank the reviewer for pointing this out. We agree that it is important to increase the visibility of this section to ensure a fair comparison between methods in future work that uses MiniHack.

### Comparison with existing benchmarks such as Atari

We also highlight the limitations of environments such as Atari that served as a motivation behind our work on MiniHack which is more complex in a multitude of different ways. While Atari includes are only fixed, singleton, and deterministic environments where RL agents already perform at a superhuman level, MiniHack features highly customisable, procedurally generated, stochastic environments in many of which agents are unable to make meaningful progress. See our [full response](https://openreview.net/forum?id=skFwlyefkWJ&noteId=kiVWJSMIGSE) for additional details.

### On game-based environments for RL research

We would also like to highlight the importance of visually simplistic environments for RL research, which allow focusing computational resources on isolated problems such as credit assignment, skill acquisition or exploration with sparse rewards, and also facilitate accessible research without the cost of processing pixel-based observations. Furthermore, many real word-applications of RL are visually simple (e.g. [Evans & Gao, 2016](https://deepmind.com/blog/article/deepmind-ai-reduces-google-data-centre-cooling-bill-40), [Cummins et al, 2020](https://github.com/facebookresearch/CompilerGym), [Sivakumar et al, 2019](https://arxiv.org/abs/1910.04054), [Mirhoseini et al, 2021](https://www.nature.com/articles/s41586-021-03544-w)). See our [full response](https://openreview.net/forum?id=skFwlyefkWJ&noteId=kiVWJSMIGSE) for additional details.

---

### Decision · Program_Chairs · 2021-07-26

**Decision:**

Accept

**Comment:**

The proposed environment to design RL problems shows to be very flexible useful. It can be a very valuable tool for the community.